**Data Availability Statement:** Data cannot be shared publicly because of an agreement made with Salud Centro. Data are available from Salud

# Publicly funded interfacility ambulance transfers for surgical and obstetrical conditions: A cross sectional analysis in an urban middle-income country setting

Paul Truche[1]*, Rachel E. NeMoyer[1], Sara Patiño-Franco[2], Juan P. Herrera-Escobar[3], Myerlandi Torres[4], Luis F. Pino[5], Gregory L. Peck[1,6]

1 Department of Surgery, Rutgers Robert Wood Johnson Medical School, New Brunswick, New Jersey, United States of America, 2 Department of Surgery, Universidad de Antioquia, Medellin, Colombia, 3 Center for Surgery and Public Health, Brigham and Women's Hospital, Boston, Massachusetts, United States of America, 4 Red de Salud del Centro, Pool Ambulancias Públicas, Cali, Colombia, 5 Department of Surgery, Universidad Del Valle, Cali, Colombia, 6 Department of Health Systems and Policy, Rutgers School of Public Health, Piscataway, New Jersey, United States of America

☉ These authors contributed equally to this work.
* Paul.truche@gmail.com

## Abstract

### Introduction

Interfacility transfers may reflect a time delay of definitive surgical care, but few studies have examined the prevalence of interfacility transfers in the urban low- and middle-income (LMIC) setting. The aim of this study was to determine the number of interfacility transfers required for surgical and obstetric conditions in an urban MIC setting to better understand access to definitive surgical care among LMIC patients.

### Methods

A retrospective analysis of public interfacility transfer records was conducted from April 2015 to April 2016 in Cali, Colombia. Data were obtained from the single municipal ambulance agency providing publicly funded ambulance transfers in the city. Interfacility transfers were defined as any patient transfer between two healthcare facilities. We identified the number of transfers for patients with surgical conditions and categorized transfers based on patient ICD-9-CM codes. We compared surgical transfers from public vs. private healthcare facilities by condition type (surgical, obstetric, nonsurgical), transferring physician specialty, and transfer acuity (code blue, emergent, urgent and nonurgent) using logistic regression.

### Results

31,659 patient transports occurred over the 13-month study period. 22250 (70.2%) of all transfers were interfacility transfers and 7777 (35%) of transfers were for patients with surgical conditions with an additional 2,244 (10.3%) for obstetric conditions. 49% (8660/17675) of interfacility transfers from public hospitals were for surgical and obstetric conditions vs 32%

Centro Ethics Committee (contact via Jhon Murillo, saludcentro@saludcentro.gov.co) for researchers who meet the criteria for access to confidential data.

**Funding:** The authors received no specific funding for this work.

**Competing interests:** The authors have declared that no competing interests exist.

(1466/4580) for private facilities (P<0.001). The most common surgical conditions requiring interfacility transfer were fractures (1,227, 5.4%), appendicitis (913, 4.1%), wounds (871, 3.9%), abdominal pain (818, 3.6%), trauma (652, 2.9%), and acute abdomen (271, 1.2%).

## Conclusion

Surgical and obstetric conditions account for nearly half of all urban interfacility ambulance transfers. The most common reasons for transfer are basic surgical conditions with public healthcare facilities transferring a greater proportion of patient with surgical conditions than private facilities. Timely access to an initial healthcare facility may not be a reliable surrogate of definitive surgical care given the substantial need for interfacility transfers.

## Introduction

Emergency and essential surgical conditions account for up to one third of the total global burden of disease [1]. Treatment of these surgical conditions requires delivery of both timely and definitive surgical care that relies heavily on out-of-hospital emergency care' (OHEC). OHEC has proven difficult to systematically examine in low- and middle-income countries (LMICs) [2]. Given this known challenge, the World Health Organization (WHO) and the World Bank advocate for the collection, analysis, and interpretation of data that informs a timely healthcare delivery integral to the achievement of definitive surgical care globally [3–5].

In 2015, the Lancet Commission on Global Surgery set a 2030 target for 80% of LMIC populations to have timely access to surgical care within 2 hours [6]. The absolute number or proportion of patients who reach a facility within 2 hours, but still do not receive definitive surgical care, is not well quantified in the LMIC setting. The "Three Delay" framework helps assess delay by categorizing delay into three types: 1) delay in deciding to seek care at a facility; 2) delay in reaching a facility; and 3) delay in the delivery of definitive care once at a facility. These delays have been proposed in maternal-fetal health frameworks to assess a continuum of delay in low resource settings [7,8]. For LMIC settings challenged by delays in the prehospital system, a more thorough understanding of interfacility transfers for surgical conditions may help to identify second and third delay factors of access to definitive surgical care.

With continued emphasis on the need for surgical system strengthening in LMICs, evaluation of interfacility transfer offers an understanding of the nuanced difference between timely arrival (2nd delay) and delivery of definitive surgical care (third delay). An interfacility transfer as a unit of analysis may represent a proxy of distinct measurement of second versus third delay, whereby patients may reach a facility (no second delay), but must be transferred to obtain the definitive surgical care they need (third delay). Each has very different targets for improvement.

In Colombia, ambulance access is covered through a national free ambulance service for all Colombian citizens with a large private ambulance sector supplementing transport for those with private insurance [9]. This provides an ideal setting to study the relationship between interfacility transfers and access to definitive surgical care. The aim of this study is to determine the number of interfacility transfers in an urban MIC setting required for surgical and obstetric conditions to better understand access to a facility that may offer definitive surgical care for the MIC population.

## Methods

A large retrospective review was performed of ambulance transports by Cali's single public ambulance agency, Salud Centro, from April 2015 through April 2016. Salud Centro provides

both emergency transports and interfacility transfers both in and around the city of Cali, Colombia. An administrative records database was queried for variables including patient gender, age, transport date, origin facility, destination facility, diagnoses code by ICD-9-CM classification, transport priority level (code blue, emergent, priority, urgent), transferring provider specialty and receiving provider specialty.

All patients who underwent an interfacility transfer during the study period were included. Due to the number of patients transported to or from a private home, or public space, we defined interfacility transfers as transfers in which both the origin and destination facility were a hospital or clinic by the presence of multiple specialties. Facilities were further categorized by public vs. private status, based on review of each facilities website or online listing, and whether they accepted publicly insured patients. All transfers initiated from a home or private physician's office, and discharges including transports with a final destination of a home residence, were excluded.

"Receiving provider specialty" was defined for each transfer by specialty and included emergency medicine, surgery, obstetrics, intensive care, internal medicine, pediatrics, or other specialties. We further sub-categorized surgeons by subspecialty including: ear-nose-and-throat (ENT), cardiothoracic (CT), gastroenterological (GI), general surgery, urology, vascular, surgical oncology, orthopedic surgery, plastic surgery, neurosurgery, ophthalmology, and pediatric surgery.

Each patient transport had an associated primary diagnosis which was used to classify patients by condition (e.g., all fracture codes were grouped as fracture). ICD-9-CM codes were grouped by author consensus with surgical conditions defined as conditions that may require care by a surgeon. Obstetric conditions were defined as all diagnoses related to pregnancy or childbirth and the related complications thereof. The study population diagnoses represented 2,273 unique ICD-9-CM codes in total.

Descriptive statistics for patient characteristics were calculated including means and standard deviations. Chi squared tests were used to compare categorical variables, and t-tests to compare continuous variables. Univariate logistic regression was used to calculate the odds of transfer origin from a public vs. private facility for each triage level and each condition and transfer priority level. Missing data was handled through complete case analysis. Statistical significance was determined at the Type I error rate of a = 0.05. All statistical analyses were performed using the SAS statistical software package version 9.4.

Institutional Review Board (IRB) approval for this study was obtained from Rutgers Robert Wood Johnson Medical School (Pro 20170000334). The Universidad de Valle shared a formal data sharing agreement with Salud Centro Ambulance service. IRB and ethics committees at both institutions waived the requirement for informed consent given all data was de-identified.

## Results

A total of 31,659 patients were transported by Cali's public ambulance system over the study period. 22250 (70.2%) transports were interfacility transfers originating from 425 different facilities. Fifty of the 425 facilities (12%) contributed to 90% of the transfers with the top 20 facilities (5%) contributing to 70% of the total transfers. Surgical conditions accounted for 35% (7,777) of interfacility transfers. Obstetric and gynecologic conditions, and nonsurgical conditions accounted for the remaining 10.2% (2349) and 55.4% (12305) respectively.

17,675 (79.4%) transfers originated from public facilities and 4,580 originated from private facilities. 49% (8660/17675) of interfacility transfers from public hosspitals were for surgical and obstetric conditions vs 32% (1466/4580) for private facilities (P<0.001). Public facilities transferred to other public facilities 61% of the time, while private facilities transferred to other private facilities, 77% of the time (p<0.0001) [Table 1].

**Table 1. Interfacility transfers.**

| Origin Hospital Type | All Transfers (n = 22,255) | | Transfers for Surgical/Obstetric Conditions (n = 9,987) | |
|---|---|---|---|---|
| | Private n (%) | Public n (%) | Private n (%) | Public n (%) |
| Number of Transfers | 4580 (21) | 17675 (79) | 1466 (15) | 8521 (85) |
| Destination Hospital | | | | |
| Private | 3511 (77) | 6676 (40) | 1025 (70) | 2610 (23) |
| Public | 1011 (13) | 10592 (60) | 417 (30) | 5730 (67) |
| Gender | | | | |
| Male | 2498 (55) | 9160 (52) | 760 (52) | 4346 (51) |
| Female | 2082 (45) | 8515 (48) | 706 (48) | 4175 (49) |
| Mean Age (years) | 41.2 (SD* 26.2) | 46.5 (SD 26.4) | 40.0 (SD 24.9) | 34.1 (SD 23.8) |

*SD: Standard Deviation.

A surgeon was the receiving specialist for 6,616 (30.1%) transfers, an internist for 6,000 transfers (27.3%), an obstetrician for 2,805 transfers (12.7%). The proportion of interfacility transfers initiated by surgeons from public facilities was higher than private facilities (33% vs. 15%; p <0.001) [Table 2].

Fourty-nine percent (10825) of interfacility transfers were emergent ("code blue" or "emergent"). The use of each triage level as an indicator of disease severity was statistically different between public and private facilities for transfer characterized as code blue (p<0.001), emergent (p<0.001), and urgent (p<0.001). Private hospitals had a higher proportion of transfers initiated as code blue and/or emergent [Table 3].

The top 21 conditions requiring interfacility transfers included many surgical diagnoses (ICD-9-CM codes), including fractures, appendicitis, wounds, abdominal pain, trauma, and labor complications [Table 4].

For patients undergoing interfacility transfer for appendicitis, the odds of the transfer originating from a public facility were 4.4 times higher than for a private facility (OR = 4.4, 95% Confidence Interval (CI) 3.3–5.8, p<0.001). For patients transferred for wounds, the odds of the transfer originating from a public facility were 7.5 times higher than for a private facility (OR = 7.5, 95% CI 5.2–10.8, p<0.001]). There were no statistical differences between origin facility type for fractures (p<0.06), abdominal pain (p<0.63), or trauma (p<0.92) [Table 5].

**Table 2. Transfers by conditions and specialty.**

| | Private n (%) | Public n (%) |
|---|---|---|
| Total | 4580 (21) | 17675 (79) |
| Transfer by Condition Type | | |
| Non-surgical | 3114 (68) | 9014 (51) |
| Obstetric | 229 (5) | 2120 (12) |
| Surgical | 1237 (27) | 6540 (37) |
| Transfer by Specialty Type | | |
| Intensive Care | 251 (5) | 1177 (7) |
| Internal Medicine | 827 (18) | 5173 (29) |
| Obstetrics | 270 (6) | 2535 (14) |
| Surgery | 707 (15) | 5909 (33) |
| Pediatrics | 225 (5) | 1801 (10) |
| Other specialty | 2300 (50) | 1080 (6) |

**Table 3. Odds of interfacility transfer for public versus private facility based on triage level.**

| Triage level | Private n (%) | Public n (%) | Odds Ratio (95% CI[1]) | p-value |
|---|---|---|---|---|
| Code blue | 701 (4) | 39 (0.8) | (44.4–229.8) | **<0.0001** |
| Emergent | 10124 (58.2) | 2056 (45.1) | (51.1–100.5) | **<0.0001** |
| Priority | 6087 (35.0) | 574 (12.6) | (17.6–21.9) | 0.119 |
| Urgent | 473 (2.7) | 1888 (41.4) | (37.1–48.3) | **<0.0001** |

[1] CI: Confidence Interval.

Bolded estimates have a significant Wald Chi-Square p-value, p<0.05.

Table totals are based on complete cases used for regression.

## Discussion

This study reveals up to one-third of patients undergoing interfacility transfers in an urban MIC setting are transferred for surgical conditions. Common surgical conditions requiring interfacility transfer from a primary healthcare facility include fractures, appendicitis, wounds,

**Table 4. Odds of interfacility transfer originating from a public versus private facility by diagnosis (reference is private).**

| Diagnosis by ICD9 Code | n | Public n (%) | Private n (%) | Odds Ratio (95% CI*) | p-value |
|---|---|---|---|---|---|
| Fracture | 1217 | 992 (89.4) | 225 (10.6) | 1.15 (0.99, 1.34) | 0.07 |
| Fever | 965 | 863 (94.2) | 102 (5.8) | 2.25 (1.83, 2.77) | **<0.0001** |
| Appendicitis | 903 | 851 (94.2) | 52 (5.8) | 4.40 (3.32, 5.84) | **<0.0001** |
| Wound | 859 | 829 (96.5) | 30 (3.5) | 7.46 (5.18, 10.76) | **<0.0001** |
| Abdominal Pain | 809 | 648 (80.0) | 161 (20.0) | 1.04 (0.88, 1.25) | 0.66 |
| Unspecified Medical Condition | 776 | 111 (14.3) | 665 (85.7) | 0.04 (0.03, 0.05) | **<0.0001** |
| MI** | 759 | 608 (80.1) | 151 (19.1) | 1.04 (0.87, 1.25) | 0.68 |
| Pneumonia | 733 | 609 (83.0) | 124 (17.0) | 1.28 (1.05, 1.56) | 0.01 |
| Cerebrovascular Disease | 676 | 525 (77.6) | 151 (22.3) | 0.90 (0.74, 1.08) | 0.25 |
| Trauma | 642 | 509 (79.3) | 133 (20.7) | 0.99 (0.82, 1.20) | 0.92 |
| Labor complication | 587 | 538 (91.7) | 49 (8.3) | 2.90 (2.16, 3.90) | **<0.0001** |
| Abortion | 499 | 470 (94.2) | 29 (5.8) | 4.29 (2.94, 6.25) | **<0.0001** |
| Infection | 436 | 346 (79.2) | 91 (20.8) | 0.98 (0.78, 1.24) | 0.9 |
| Hypertension | 424 | 351 (82.8) | 73 (17.2) | 1.25 (0.97, 1.61) | 0.09 |
| Chest Pain | 368 | 324 (88) | 44 (12) | 1.93 (1.40, 2.08) | **<0.0001** |
| Seizure | 361 | 309 (85.6) | 52 (14.4) | 1.55 (1.15, 2.08) | 0.0031 |
| GI# bleed | 311 | 257 (82.6) | 54 (17.4) | 1.24 (0.92, 1.66) | 0.18 |
| Labor | 311 | 280 (90) | 31 (10) | 2.36 (1.63, 3.43) | **<0.0001** |
| CHF+ | 309 | 256 (82.9) | 53 (17.1) | 1.26 (0.93, 1.69) | 0.14 |
| COPD++ | 304 | 233 (76.6) | 71 (23.4) | 0.85 (0.65, 1.11) | 0.23 |
| Acute Abdomen | 270 | 239 (88.5) | 31 (11.5) | 0.49 (0.34–0.73) | 0.0001 |

*CI: Confidence Interval;

**MI: Myocardial Infarction;

#GI: Gastrointestinal bleed;

+CHF: Congestive Heart Failure;

++COPD: Chronic Obstructive Pulmonary Disease.

Bolded estimates have a significant Wald Chi-Square p-value, p<0.05.

Totals are based on complete cases used for regression.

**Table 5. Interfacility transfers for common surgical conditions.**

| Surgical Condition | n (%) | Odds Ratio (95% CI*) | p-value |
|---|---|---|---|
| Fracture | 1,227 (5.4) | 1.15 (0.99, 1.34) | 0.07 |
| Appendicitis | 913 (4.1) | 4.40 (3.32, 5.84) | **<0.0001** |
| Wound | 871 (3.9) | 7.46 (5.18, 10.76) | **<0.0001** |
| Abdominal Pain | 818 (3.6) | 1.04 (0.88, 1.25) | 0.66 |
| Trauma | 652 (2.9) | 0.99 (0.82, 1.20) | 0.92 |

Odds ratio represent odds of transfer being initiated from a public hospital compared to a private hospital among patients with each condition. Bolded estimates have a significant Wald Chi-Square p-value, p<0.05, *CI: Confidence Interval.

abdominal pain, and trauma. When combined, obstetric and surgical conditions make up nearly half of all interfacility transfers. These results reveal a previously unrecognized, and significant, proportion of interfacility transfers that occur due to definitive surgical care being sought at a second healthcare facility in an urban MIC setting.

Prehospital triage systems rely on the development and implementation of the six WHO health system building blocks [10]. As it pertains to delay of surgical care and prehospital triage, Infrastructure and Workforce are two of six building blocks requiring strengthening according to the Lancet Commission on Global Surgery (LCOGS) within a National surgical, obstetric and anesthesia plan (NSOAP) [11,12]. Interfacility transfer is a specific example of prehospital triage within a system's Infrastructure. Longer transport times have been associated with worse outcomes and occur as a result of poor Infrastructure and lack of robust health systems [13–16]. In recognition of how difficult it has been for both low- and middle-income countries to formally integrate the prehospital system with hospital level healthcare service delivery and outcomes, calls for action toward important progress in health system strengthening have been made [17–19]. From the examples above, access to surgical care in LMICs remains a complex interprofessional agenda still needing further multi-agency development and implementation, as well as specific targets of collaborative evaluation [19–22]. In Latin America, where the majority of countries are middle income (MIC), prehospital trauma triage avoids time delays to favorably impact outcomes once definitive care is obtained [23,24]. For example, in Colombia the majority of ambulances are privately owned and staffed, while the *Servicio de Atención Médica de Urgencia* (SAMU) is the federal subsidized ambulance service for all Colombian citizens [9]. Quality has been found to be comparable to most higher income countries and is perhaps related to ambulance one hour availability nearly 100% of the time in Colombia [10]. However, large studies suggest Latin American countries may have some of the highest interfacility transfers in the world with an unknown impact on definitive surgical time [25].

A number of studies have utilized time to arrival to the nearest hospital i.e., first hospital, as representative of "timely access" to surgical care in LMICs [26,27]. To date, in the largest study of interfacility transfer and surgical disease to exist, Latin America had the highest delay of fracture care and the highest proportion of interfacility transfers [25,28]. There were concerning delays in admission to a hospital for fractures in the region, with 88·7% of open fracture, and 44·7% of closed fracture admissions delayed], with a significant association between such delays and interfacility transfer [25]. However, it is unknown whether interfacility transfers are a cause or outcome of second (reaching care) or a third (receiving care) delay. Due to this dual impact, rates of interfacility transfers could be useful data for improved modeling of definitive access to timely surgery in MICs, where absolute times of facility arrival, and definitive surgical intervention data may not be integrated.

Our findings suggest that "timely access" to a "first" healthcare facility is not a reliable surrogate of definitive surgical care given the substantial need of interfacility transfer. Modeling of surgical access has predominantly relied on estimates of patients' ability to reach a hospital, and do not account for the possibility of needing an interfacility transfer for definitive surgical healthcare at a second healthcare facility [26–29]. Thus, models that exclude this large proportion of patients who undergo transfer, overestimates global "timely access" to definitive surgical healthcare.

Low-income country evidence has shown a survival benefit for trauma and obstetrical patients when interfacility transfer is more readily available [17,30,31]. When the entire spectrum of interfacility transfers is considered, including modifying factors of initial pre-hospital system triage that could potentially circumvent the need for interfacility transfer altogether, or improving the efficiency and effectiveness of appropriate interfacility transfers when needed, the actual cause or effect of second and third delays can be more precisely assessed and addressed. As evidenced by the absolute number and proportion of patients in this study requiring interfacility transfers for surgical conditions, a large population of patients stand to benefit from these surgical system preparedness and service delivery-based considerations crucial to Infrastructure and Workforce strengthening.

The factors of delay in timely and quality surgical care delivery represent tangible targets for evidence-based policy improvement of access to definitive surgical healthcare. Although universal health insurance in Colombia has expanded the coverage of healthcare services, timely and quality delivery of definitive healthcare services is thought to vary among socioeconomic groups and the regional rural-urban geographic divide [32,33]. The Organization for Economic Co-operation and Development (OECD) reported that *vertical integration* between providers and the over ninety-nine public and private health insurers in Colombia has allegedly contributed to a fragmented healthcare delivery, reduced competition at the provider level, making it difficult to allocate sparse human resources equitably and efficiently [34]. This vertical integration, whereby healthcare delivery entities may be entwined with individual insurers, makes a surgical healthcare system that is clinical outcomes based, difficult to assess and achieve when cooperation is complicated among competitive, cross-sectoral healthcare facilities and markets. The rate of interfacility transfers from our data may corroborate the concerns engendered by the OECD's statement, as definitive delivery of surgical healthcare requires specialized Infrastructure and Workforce that may not be equitably expanded across the public and private hospitals within the local healthcare system.

In our study both public and private hospitals utilize the public sector Emergency Medical Services (EMS) for interfacility transfers. When considering a common condition such as appendicitis, our data demonstrates that public hospitals have a higher proportion of interfacility transfers for appendicitis when compared to private. The disproportionate rate of public-private interfacility transfers may reflect the imbalance inherent with vertical integration where private hospitals are equipped to deliver definitive surgical healthcare via sustainable Infrastructure and Workforce. Studies in LMICs are replete with respect to the imbalance in healthcare Infrastructure and Workforce between public and private sectors and the inequitable impact on sub-populations [35–37].

The fact that the most common conditions being transferred by the public sector in our study were largely surgical may reflect a system that varies in Infrastructure and Workforce capacity key to delivering both timely *and* definitive surgical healthcare. Interfacility transfer, therefore, must best considered as a solution to the *vertical integration* allegedly present in Colombia [33]. Further data surrounding the interface between vertically independent healthcare entities is needed to understand how to best utilize the Infrastructure and surgical Workforce for the population's need, especially within competitive markets. Analysis of this data

could then better direct prehospital and interfacility transfer policies in the MIC towards a clinical outcomes and equity driven delivery of *definitive* surgical care.

When we consider the private sector and interfacility transfer, surgical conditions requiring transfer by private hospitals were more likely to be of the highest urgency level (code blue). This raises the question of whether private hospitals tend to use the publicly funded ambulance system for higher acuity patients. Nevertheless, both public and private sector examples in our study represent a cross-sectoral need for policy development that establishes an equitable delivery of timely, definitive healthcare in this region. Prioritization of definitive delivery of healthcare is especially important given the large population burden herein that requires interfacility transfer for treatment of emergency and essential surgical and obstetric conditions.

## Limitations

Our study examines only interfacility transfers executed through the public ambulance sector and does not account for transfers provided by the other privately-owned ambulance companies in Cali. The analyses conducted are based exclusively on administrative data that is limited to a single public EMS agency and does not include clinical variables other than diagnoses codes from hospitals. Clinical status of patients undergoing transfer were limited to the triage level initiated at the time of transfer. The clinical outcomes of transferred patients were also not evaluated, and merging data from local governance, finance, infrastructure, workforce, and service delivery stakeholders will be critical for this in the future. Although ICD-9-CM codes were used in our data analysis, local billing practices have an unknown impact on codes chosen and recorded for individual patients. The above are clear examples of measurement and information bias.

Although some hospitals utilizing the public EMS agency were utilized in our analysis, this study does not account for the private ambulances operated by individual hospitals. A selection bias of this magnitude requires a large collaborative cross-agency study to overcome these limitations in study design. Finally, the formal and informal transfer agreements motivated by contractual commitments between primary and secondary healthcare facilities and the public EMS agency, as well as the potential non-clinical, administrative, or confounding variables associated with reasons why the secondary hospitals are accepting the volume of interfacility transfers exhibited, were not controlled for in this study.

## Conclusion

Surgical and obstetric conditions account for nearly half of all interfacility ambulance transfers in the urban MIC setting. The most common reasons for transfer are basic surgical conditions, with public healthcare facilities contributing a greater proportion of interfacility transfers for surgical conditions. Timely access to an initial healthcare facility may not be a reliable surrogate of definitive surgical care given the substantial need for interfacility transfers. Further research towards the nature of inter-hospital interface through prehospital transfer is needed to understand how to best utilize the infrastructure and surgical workforce in MICs.

## Author Contributions

**Conceptualization:** Paul Truche, Rachel E. NeMoyer, Sara Patiño-Franco, Juan P. Herrera-Escobar, Myerlandi Torres, Luis F. Pino, Gregory L. Peck.

**Data curation:** Paul Truche, Rachel E. NeMoyer, Juan P. Herrera-Escobar, Myerlandi Torres, Gregory L. Peck.

**Formal analysis:** Paul Truche, Rachel E. NeMoyer.

**Investigation:** Paul Truche, Gregory L. Peck.

**Methodology:** Paul Truche.

**Project administration:** Paul Truche, Rachel E. NeMoyer.

**Resources:** Paul Truche.

**Supervision:** Rachel E. NeMoyer.

**Writing – original draft:** Paul Truche, Rachel E. NeMoyer, Sara Patiño-Franco.

**Writing – review & editing:** Paul Truche, Rachel E. NeMoyer, Sara Patiño-Franco, Juan P. Herrera-Escobar, Myerlandi Torres, Luis F. Pino, Gregory L. Peck.

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
