## [Editor Report · Decision Letter 0]

23 Apr 2020

PONE-D-20-00032

Publicly Funded Interfacility Ambulance Transfers for Surgical and Obstetrical Conditions: A Cross Sectional Analysis in the Urban Middle-Income Country Setting

PLOS ONE

Dear Dr NeMoyer,

Thank you for submitting your manuscript to PLOS ONE. After careful consideration, we feel that it has merit but does not fully meet PLOS ONE’s publication criteria as it currently stands. Therefore, we invite you to submit a revised version of the manuscript that addresses the points raised during the review process.

We would appreciate receiving your revised manuscript by Jun 07 2020 11:59PM. To enhance the reproducibility of your results, we recommend that if applicable you deposit your laboratory protocols in protocols.io, where a protocol can be assigned its own identifier (DOI) such that it can be cited independently in the future. For instructions see: http://journals.plos.org/plosone/s/submission-guidelines#loc-laboratory-protocols

We look forward to receiving your revised manuscript.

Kind regards,

Denis Bourgeois

Academic Editor

PLOS ONE

Journal Requirements:

2. In the ethics statement in the manuscript and in the online submission form, please provide additional information about the patient records used in your retrospective study. Specifically, please ensure that you have discussed whether all data were fully anonymized before you accessed them and/or whether the IRB or ethics committee waived the requirement for informed consent. If patients provided informed written consent to have data from their medical records used in research, please include this information.

3. During your revisions, please note that a simple title correction is required to ensure the title is written in standard English: "Publicly Funded Interfacility Ambulance Transfers for Surgical and Obstetrical Conditions: A Cross Sectional Analysis in an Urban Middle-Income Country Setting". Please ensure this is updated in the manuscript file and the online submission information.

Additional Editor Comments (if provided):

Please see minor comments in the attachment
---

## [Author Response · Author response to Decision Letter 0]

26 Jun 2020

We are excited to resubmit our manuscript following the proposed changes. We have

addressed the reviewer comments as well as made some minor changes to the structure of the

discussion in order to improve clarity and readability. We have highlighted areas of the

manuscript with major changes in yellow.

We addressed the following specific reviewer comments:

1) Abstract, Introduction: LMIC was spelled out (low- and middle-income (LMIC))

2) Introduction, Line 62: Wording changed as recommended (low and middle income

countries (LMICs) population)

3) Introduction, Line 70-72: Three lines were added specific to Colombia, “In Colombia,

ambulance access is covered through a national free ambulance service for all

Colombian citizens with a large private ambulance sector supplementing transport for

those with private insurance. 9 This provide an ideal setting to study the relationship

between interfacility transfers and access to surgical care.”

4) Methods, Line 125: IRB protocol number was added from Robert Wood Johnson (Pro

20170000334)

5) Results, Table 1 and 2 were formatted to be easier to interpret

6) Discussion, Line 213: We added in a recent reference regarding Colombia and delays

related to fracture care, “In a recent paper, Latin America (including Venezuela and

Mexico) had the highest delay of fracture care and the highest proportion of interfacility

transfers. There were concerning delays in admission to a hospital in the region [173

(88·7%) with open fractures and 426 (44·7%) with closed fractures] and a

significant association between such delays and regional interfacility transfer.”

7) Discussion: We have included reference to the WHO and guided part of the discussion

towards how countries can work to adapt research to improving access to surgical care

through prehospital care.

---

## [Editor Report · Decision Letter 1]

19 Oct 2020

Publicly Funded Interfacility Ambulance Transfers for Surgical and Obstetrical Conditions: A Cross Sectional Analysis in an Urban Middle-Income Country Setting

PONE-D-20-00032R1

Dear Dr. NeMoyer,

We’re pleased to inform you that your manuscript has been judged scientifically suitable for publication and will be formally accepted for publication once it meets all outstanding technical requirements.

Kind regards,

Junaid Ahmad Bhatti

Academic Editor

PLOS ONE
---

## [Editor Report · Acceptance letter]

26 Oct 2020

PONE-D-20-00032R1 

Publicly Funded Interfacility Ambulance Transfers for Surgical and Obstetrical Conditions: A Cross Sectional Analysis in an Urban Middle-Income Country Setting 

Dear Dr. NeMoyer:

I'm pleased to inform you that your manuscript has been deemed suitable for publication in PLOS ONE. Congratulations! Your manuscript is now with our production department. 

Kind regards, 

on behalf of

Dr. Junaid Ahmad Bhatti 

Academic Editor

PLOS ONE